# Preparation and Characterization of Polyethersulfone-Ultrafiltration Membrane Blended with Terbium-Doped Cerium Magnesium Aluminate: Analysis of Fouling Behavior

**DOI:** 10.3390/molecules28062688

**Published:** 2023-03-16

**Authors:** Gouled Aouled, Saleem Raza, Ehsan Ghasali, Asif Hayat, Yasin Orooji

**Affiliations:** College of Geography and Environmental Sciences, Zhejiang Normal University, Jinhua 321004, China

**Keywords:** Polyethersulfone, ultrafiltration membrane, terbium-doped cerium magnesium aluminate (Ce_0.67_Tb_0.33_MgAl_11_O_19_), blending, characterization, fouling behavior

## Abstract

In this study, various techniques, including X-ray diffraction (XRD), high-resolution transmission electron microscopy (HRTEM), scanning electron microscopy (SEM), energy-dispersive X-ray spectrometry (EDS) mapping, X-ray photoelectron spectroscopy (XPS), and water-contact-angle goniometry (WCAG), were used to characterize the crystalline structure and morphological properties of terbium-doped cerium magnesium aluminate (Ce_0.67_Tb_0.33_MgA_l1_1O_19_ or CMAT) in powder form. The results demonstrated that CMAT was successfully synthesized with a particle size of less than 5 µm and a fully evident distribution of elements, as revealed by the SEM images. This was further confirmed by the XRD and HRTEM images. XPS analysis confirmed the presence of all necessary components in CMAT. Additionally, WCAG results showed that the contact angle of CMAT was more hydrophilic with a value of 8.4°. To evaluate its performance, CMAT particles were dispersed in a Polyethersulfone (PES) solution and used to modify a PES ultrafiltration membrane through a phase-inversion method. The resulting membranes were characterized by SEM, atomic force microscopy (AFM), thermogravimetric analysis (TGA), WCAG, and permeability performance and fouling experiments. The addition of CMAT to the PES membranes did not have a significant effect on the structure of the SEM images of the top layer and cross-section of surface properties. However, increasing the concentration of CMAT improved the membrane surface roughness in AFM, and the modified membranes had the ability to resist fouling. The addition of CMAT did not lead to significant energy loss, indicating that the heat flux loss observed can indeed be explained by the amount of C-OH on the PES membrane’s surface. The contact angle of the membranes became more hydrophilic with increasing concentration of CMAT from PES G0 to PES G7. The PES origin membrane showed a higher permeation than the membranes mixed with CMAT, and the modified membranes with CMAT displayed significant fouling resistance.

## 1. Introduction

The technique of membrane separation has developed into a viable alternative for performing separations and purifications in a variety of contexts [1]. The rising desire for a technology that is both cost effective and favorable to the environment, as well as the flexibility and performance dependability of membrane systems, are all factors that contribute to the numerous benefits that membrane technology offers [2,3]. Ultrafiltration is a significant membrane separation technique that has developed into a prominent separation tool for a wide variety of industrial applications owing to its one-of-a-kind separation capabilities and low amount of required energy. The application areas of ultrafiltration include the processing of food and chemicals and treatment [4,5]. The strong chemical resistance, good thermal stability, oxidation resistance, and mechanical qualities of Polyethersulfone (PES) membranes have led to their widespread use in a variety of industries, including water treatment, food processing, and biomedical research [6,7]. The hydrophobic characteristic of PES, on the other hand, often leads to fouling of the pure PES membrane used in water treatment [8].

Due to the comparatively low hydrophilicity of PES membranes, they experience membrane fouling in spite of these benefits [9]. One of the main issues with MF and UF operation is fouling. This undesirable event adds to the reduction in the permeability of membranes or possibly their degradation, raising the expenses of the operation [10]. Researchers are now looking for contemporary ways to enhance the antifouling capabilities of membranes [11]. Modifying membranes using nanoparticles seems to be one of several methods being attempted to reduce fouling. Numerous modifying substances have been suggested, such as one-dimensional carbon nanotubes (CNTs) or zero-dimensional TiO_2_ nanoparticles (NPs), halloysite nanotubes (HNTs), and titania/titanate nanotubes (TNTs) [12,13]. Additionally, several metallic nanoparticle kinds, including Ag and Cu, metal oxides including iron oxides, ZrO_2_, Al_2_O_3_, ZnO, etc., and/or their composites, have been studied throughout time [14,15,16]. Incorporating these nanoparticles was discovered to generally enhance membrane hydrophilicity, permeability, and biofouling resistance [17]. Since the invention of the membrane system, membrane fouling has continuously been a barrier that limits the output of the purification process by worsening the amount (permeation flux) and quality of the acquired water as well as the process’s energy consumption [18]. It is important to keep in mind that the fouling phenomena in the membrane system is a global issue since the membrane comes into direct touch with many types of foulants in water [19,20]. A significant challenge is the presence of fouling in many applications, including industrial and municipal. The many types of fouling, which impact all membrane processes used in water purification, include silting, scaling, organic fouling, and biofouling [21,22]. Using model organic foulants to investigate the fouling behavior is a common method employed in microfiltration (MF), ultrafiltration (UF), nanofiltration (NF), and reverse osmosis (RO) [23].

Ultrafiltration UF membranes are routinely implemented in the purification of proteins for pharmaceutical and biotechnological needs. During operation, the proteins gradually foul the membranes. The prevention of fouling or elimination of the fouling layer is needed to keep a good protein yield and a high membrane selectivity. Due to these problems, several studies have been conducted in order to enhance PES membrane structure and performance [24,25]. Membrane fouling is one the main disadvantages of the pressure-driven membrane processes applied to wastewater treatment. Membrane fouling can be caused by mineral precipitation, the attachment of colloids and dissolved organics and growth of microbes on the membrane surface [26]. Membrane fouling due to the adsorption of organic pollutants reduces water transport across the membrane and deteriorates other functional properties of the membrane surfaces, sequentially increasing energy consumption and shortening membrane life [27]. Due to its several beneficial physicochemical properties, including its ease of processing, strong heat-aging resistance, and environmental strength, Polyethersulfone (PES) has recently become a significant membrane separation material [28]. However, because of its structure, PES has a natural hydrophobicity that results in a low membrane flow and poor antifouling characteristics [29,30]. As a result, attempts have been made to modify PES either chemically or physically in order to enhance its qualities. For instance, many techniques have been examined in order to increase the hydrophilicity of PES membranes [28].

In addition, the terbium-doped cerium magnesium aluminate (CMAT), which has a magneto plumbite structure, has tremendous applications as the green-emitting component in trichromatic lamps, plasma display panels (PDP), and very-high-loading and long-lifetime fluorescent lamps. Because of its high quantum efficiency, thermal stability, and high durability against intense UV radiation, CMAT has a magneto plum (FL) [31,32]. The shape, size, and distribution of the particle which are heavily influenced by the synthesis method as well as the composition, crystal type, and phase of phosphors are known to affect its chemical and physical characteristics [33,34]. Phosphor particles should have a precise spherical form and a restricted size distribution, and not aggregate. A phosphor’s spherical form is ideal for high brightness and high resolution because of its high packing density and minimal light scattering [35,36]. Commercial phosphors have been produced on a wide scale using the solid-state approach, although this process has limitations when it comes to morphological control [37].

In the present study, CMAT was used as a hydrophilic polymeric material to integrate with Polyethersulfone (PES) matrix membranes. The original membrane was created by casting the PES polymer solution without any alterations, while CMAT material was blended into the PES solution to create seven different membranes, designated as PES G1, PES G2, PES G3, PES G4, PES G5, PES G6, and PES G7, as shown in Figure 1. The aim of this research was to synthesize CMAT phosphor with diverse morphologies through simple high-temperature methods and to characterize the terbium-doped cerium magnesium aluminate and the CMAT–PES blended ultrafiltration membranes while studying the fouling behavior of the membranes. To the best of our knowledge, this is the first study on the preparation of blended PES ultrafiltration membranes using terbium-doped cerium magnesium aluminate.

## 2. Results and Discussion

### 2.1. Characterizations of CMAT

Terbium-doped cerium magnesium aluminate with the chemical formula Ce_0.67_Tb_0.33_MgAl_11_O_19_ (CMAT) was characterized by XRD, HRTEM, SEM, XPS, and WCAG. Figure 1 illustrates the findings of an X-ray phase analysis of CMAT oxide powder samples after the sintering and grinding process. Regarding Figure 1, the dominant peak is related to the magnesium cerium terbium aluminum oxide phase with ICDD card number of 00-036-0073. The sintering process using CMAT powder led to a 6- to 15-fold increase in the size of the coherent scattering region (CSR). It seems that the synthesis process successfully led to the formation of CMAT with minimum reaction products as detection limits of XRD analyses.

A transmission electron microscopy (TEM) image of the terbium-doped cerium magnesium aluminate is shown in Figure 2. The powder was formed of separate, disproportionate particles, and the transmission electron microscopy (TEM) imaging obtained did not contain any particle agglomerates. Moreover, Figure 2 revealed the HRTEM micrographs as well as digital SAEDP using a fast Fourier transform with an inverse FFT image of the area and intensity profile–lattice distance analysis based on Gatan Digital Micrograph software at low and high magnification. Considering Figure 2, the obtained lattice distance of 0.25 nm was calculated as crystallographic orientation in (114) planes, which is in full agreement with the presented data in XRD patterns.

The Figure 3 shows a scanning electron microscopy (SEM) image of the CMAT particles, along with elemental mapping for Al, O, Mg, Ce, and Tb. The primary particles of the nanopowder are spherical in shape, as seen in Figure 3a,b. The EDX mappings presented in Figure 3c–h clearly show the presence of each component of CMAT in the powder form, which is consistent with the information provided by the XRD and TEM images, thus confirming the successful synthesis of the powder with uniformly distributed components. The laser diffraction observations indicate that the structural components of the terbium-doped cerium magnesium aluminate powder have a size range of less than 5 μm. The differences in the average particle size data obtained using different methods are due to the deviation from the spherical particle shape and the coagulation of primary powder particles, which are also evident in the SEM results.

To understand more concerning the surface-chemical bonding and chemical-valence state of the individual materials in the original green phosphor, XPS measurements were taken as shown in Figure 4. The complete XPS survey scan seen in Figure 4a reveals that the original green phosphor contains Ce, Tb, Al, Mg, and O components and the usual spectrum of C 1s 292.90 eV. In Figure 4b–g below, the sweeping of the XPS spectra for Tb3d, O1s, Mg1s, Ce3d, C1s, and Al2p illustrates adjustments in their chemical bonding. As previously indicated, the component Tb is mostly present in the original phosphor as Tb3d. As noted in Figure 4b, the binding energy for the original phosphor presents four Tb3d peaks, two peaks at 1277.2 eV and 1242.6 eV assigned to Tb 3d3/2 and two additional peaks at 1228.3 eV and 1221.7 eV attributed to Tb 3d5/2, respectively. The O 1s spectrum was fitted by three peaks, with the first peak located at a lower binding energy of 530.4 eV, the second peak at the same altitude as 531.8 eV, and the small peak, which may be assigned to the oxygen atoms of adsorbed water with a characteristic binding energy of approximately 532.5 eV [38], attributed to C-O, O-C, and O-C, correspondingly (shown in Figure 4c). Furthermore, the binding energy of magnesium also appears in Figure 4d, in which there is just one peak at 1303.8 eV that is attributable to Mg-O. Figure 4e demonstrates the values of Ce3d’s binding energies in three peaks, one peak at 905.1 eV attributed to Ce 3d_3/2_ and two peaks appearing at 886.2eV 882.2eV attributed to Ce 3d_5/2_, respectively. The carbon-binding energy is additionally presented in Figure 4f, which indicates two peaks at 284.8 eV and 288.1 eV related to C-C and C=O and two peaks at 74.0 eV and 74.9 eV that have been associated with the Al-O binding energy; the aluminum-binding energy is illustrated in Figure 4g. Overall, the XPS study shows satisfactory results and provides evidence that all the elements are present in the (CMAT) materials.

The water-contact-angle goniometer (WCAG) at CMAT was used to measure two random surfaces of the material to avoid experimental errors. The first measurement of the contact angle of CMAT showed 5.5° (Figure 5a), and the second measurement showed 11.2° (Figure 5b). After calculating the average of these two measurements, we can conclude that CMAT is purely hydrophilic, according to Table 1.

### 2.2. Characterizations of Membranes

Appendix A shows SEM images of the top-layer surfaces and cross-sections of the PES membranes (from G0 to G7) created with varying concentrations of CMAT using ultrafiltration techniques. The images of the top-layer surfaces indicate that there were no clearly observable variations in the top surface of the PES membranes, which exhibited a flat and smooth surface, implying that the top-surface structures were not affected by the addition of CMAT. Additionally, compared to the original PES membrane, the cross-section of multiple membranes showed significantly wider pore channels, while the bottom surfaces had enlarged pores. The macroporous substructures and enlarged surface macrovoids were visually evident in the SEM images. The stronger hydrophilic qualities of CMAT could be interpreted as an effect that leads to a better demixing process.

AFM was utilized to investigate the surface properties of various membranes with different CMAT concentrations. Figure 6 illustrates three-dimensional Atomic Force Microscopy (AFM) images of the membrane surfaces. The highest points of the membrane surfaces are shown as the brightest areas, while valleys or membrane gaps are depicted as the darkest areas. It can be observed that as the concentration of CMAT increases, the modified membranes exhibit a more abundant modular structure on their surface, while the PES original G0 membrane and PES G5 with lower CMAT concentrations have smoother surfaces. It is well known that a membrane with smoother surfaces has greater fouling resistance capability [12]. We have concluded that increasing the concentration of CMAT results in an increase in membrane-surface roughness due to its role in the phase-inversion process. Moreover, we conclude that the modified membranes exhibit significant fouling resistance.

Furthermore, Appendix A in the Supporting Information shows the typical TGA graphs of the PES membranes from G0 to G7. It can be observed that there is a consistent temperature drop throughout all the TGA curves. The membranes PES G1, PES G2, PES G3, PES G4, PES G6, and PES G7 show the same endothermic peak at 80 °C and the same exothermic peaks at 40 °C. On the other hand, the G5 membrane shows an endothermic peak around 100 °C and two exothermic peaks at 40 °C and 100 °C. This means that the TGA reaction for the PES G5 membrane starts at 40 °C and ends at 100 °C, while for the other membranes, it starts at 40 °C. The addition of CMAT did not result in a significant loss of energy, which suggests that the slight temperature loss can be attributed to the presence of C-OH on the surface of the PES membrane [39].

In addition, Figure 7 illustrates the results of the water-contact angle measured on the surface of PES membranes from the original PES membrane G0 to the modified membranes (PES G1 to PES G7). Higher angles denote stronger hydrophobicity, and the contact-angle measurements can aid in subjectively evaluating the wettability of each layer. The link between contact angle and terms such as hydrophobic and hydrophilic is that contact angles near 90° or greater suggest substantial hydrophobicity, whereas lower contact angles imply increasing hydrophilicity.

The PES original membrane shows a contact angle of 71°, which means the angle is close to 90°, and the surface cannot be wetted. It naturally repels water, causing a droplet on the surface of the membrane with low surface energy, making it hydrophobic. Compared to PES G5, PES G6, and PES G7, the surface energy is high and wettable, and water spreads across, maximizing contact with the membrane surface because of their low angle, making them hydrophilic. The water-contact angle dropped from 71° for PES G0 to 26° for PES G7, decreasing as the CMAT content increased. Therefore, we can conclude that the membranes blended with a high concentration of CMAT show more hydrophilicity than the membranes with a low concentration, as shown in Figure 7.

### 2.3. Filtration Performance and Performance against BSA Antifouling

Filtration studies were carried out to investigate the flux and rejection of various membranes under a pressure of 0.5 MPa. The lowest value of pure water permeation displayed was 1 L·m^2^·h^−1^, and the maximum value was 14 L·m^2^·h^−1^ for the six graphs. Figure 8 shows the filtration of membranes as a function of time. We can observe that the permeation of water through the membranes decreased slightly from the first graph to the sixth graph. The PES origin membranes G0 and those with low concentrations of CMAT, such as PES G1 and PES G2, showed a higher permeation of water, while the membranes with higher concentrations of CMAT illustrated very low permeation. This reveals that the hydrophilicity of the membranes was greatly improved by the integration of CMAT, which also matched the findings of the contact-angle test shown in Table 1.

The deterioration of permeability and selectivity due to membrane fouling is a major issue in protein separation by membrane filtration. The rejection performance of the membranes was evaluated using BSA in the same dead-end filtration cell. The rejection of the prepared membranes was evaluated using a 0.5 g/L BSA solution. Figure 9 shows that from G0 to G7, the rejection of the membranes decreased sharply in all six graphs when the feed liquid was changed to a BSA solution due to the formation of a filtration cake by the deposition and adsorption of protein on the membrane surface. We can safely assume that the modified membranes with CMAT demonstrate significant fouling resistance.

## 3. Materials and Methods

### 3.1. Materials

Terbium-doped cerium magnesium aluminate (CMAT) was purchased from Sigma-Aldrich, and CeO_2_ (Sigma-Aldrich, 211575, powder, <5 μm, 99.9% trace metals basis), Tb_4_O_7_ (Sigma-Aldrich, 253952), MgO (Sigma-Aldrich, 342793, ≥99% trace metals basis, −325 mesh), and Al_2_O_3_ (Sigma-Aldrich, 229423 powder, 99.99% trace metals basis) were all bought from the same source Sigma-Aldrich, Michigan, MI, USA. A N-Methyl-2-pyrrolidone (NMP) reagent was used without further purification. Polyethersulfone (PES) ultrason E6020P, Polyvinylpyrrolidone (PVP) with the CAS number 9003-39-8 were purchased from Shanghai, China and Bovine serum albumin (BSA, Phygene Biotechnology Co, Ltd., #99 Gujing MaBao, High-Tech Zone Fuzhou, 350001, China). The heat-collecting magnetic stirrer, electronic scales, electric oven with constant temperature, blast, reagent bottle, glass bottle with blue cap and high-temperature sterilization thread mouth, high-borosilicate transparent flat membrane laboratory, and hair dryer used to dry the flat membrane when casting the solution were all used as received. All aqueous solutions were prepared with ultrapure water and the chemicals were of the highest purity since they were used without further purification.

### 3.2. Synthesis of the Terbium-Doped Cerium Magnesium Aluminate

For the synthesis of terbium-doped cerium magnesium aluminate (CMAT) with the chemical formula Ce_0.67_Tb_0.33_MgAl_11_O_19_, the Stoichiometric amounts of CeO_2_, Tb_4_O_7_, MgO, and Al_2_O_3_ were mixed through a high-energy mixer mill for 12 h with zirconia cups and balls (ball-to-powders ratio was 10:1) in ethanol media at a rotation speed of 285 rpm. After the mixing process, the obtained mixture was dried in a drier oven at 80 °C. To improve the proper synthesized reaction of CMAT, a pre-shaping process was performed on the mixture of powders with a uniaxial press at a pressure of 120 MPa, which led to the preparation of cylindrical green samples. A sintering process was conducted on the green bodies with a heating rate of 5 °C/min at a maximum temperature of 1200 °C with a 5 h holding time. It is worth mentioning that the sintering process also included three stages of holding time (1 h) before reaching the maximum temperature (1200 °C) at temperatures of 600 °C, 800 °C, and 1000 °C in one cycle of heating. The obtained bulk-sintered samples were grinded and sieved to obtain a uniform distribution of powders [40].

### 3.3. Preparation of Membranes

Initially, solutions of Polyvinylpyrrolidone (PVP), N-methyl-2pyrrolidone (NMP), and Polyethersulfone (PES) were prepared and separated into eight solutions. Seven of these eight solutions were then combined and mixed with different concentrations of CMAT. The resulting solutions were evenly cast onto a glass substrate using a 200 mm-thick hand-casting knife. They were then immersed in a bath at a temperature ranging from 10 °C to 20 °C for 24 h. The phase inversion occurred 10 s after casting and before the solution was transferred to the water coagulation bath. After 24 h, the membranes were removed from the bath. This process produced eight different types of PES membranes with varying concentrations of CMAT, labeled as PESG0, PESG1, PESG2, PESG3, PESG4, PESG5, PESG6, and PESG7. These membranes were stored in separate boxes for characterization and application purposes. The phase-inversion technique was used to produce both the original PES membrane and the blended PES membranes (Table 2).

### 3.4. Characterization of Terbium-Doped Cerium Magnesium Aluminate (CMAT)

The crystal structure of CMAT was analyzed using X-ray diffraction (XRD) with a Rigaku Smartlab 9 KW instrument and Cu Kaα radiation, Tokyo, Japan. The XRD scan was performed over a range of 1–90° with an increment of 0.5°. To determine the particle size and morphology of the CMAT, transmission electron microscopy (TEM) was conducted using an FEI Tencai 20 microscope at a voltage of 60 kV and scanning electron microscopy (SEM, Hitachi S-4800, Kawasaki, Japan). X-ray photoelectron spectroscopy (XPS) was performed using a Thermo Scientific ESCALAB 250Xi instrument, (Arizona, AZ, USA) with a base pressure of 4.5 × 10^−10^ MPa. The contact angle of the CMAT was measured by compressing the test sample into a solid slice with a smooth surface.

### 3.5. Characterization of Membranes

To investigate the surface morphology of the membrane, samples were first dried, cut into small pieces using scissors, and attached to the SEM plate using tweezers. A thin layer of gold was then sprayed onto the samples to provide electrical conductivity. The samples were analyzed using SEM operating at 5 kV, and pictures were captured at different magnifications. The surface morphology and roughness of the membranes, an atomic force microscope (AFM, Bruker Nanoscope, Rheinstetten, Germany) outfitted with an alpha300 was used for analysis. After preparation, the membranes were cut into smaller pieces and affixed to glass substrates. Scans of 10 × 10 μm were used to evaluate the surface roughness of the membranes. The thermal properties of the membranes were analyzed using thermogravimetric analysis with a PerkinElmer DSC7TGA instrument, Waltham, Massachusetts, USA. The surface wettability of the membranes was analyzed using a drop-shape analysis system (Easy drop by KRUSS Gmbh) and the sessile drop method. In this method, Milli-Q ultrapure water purification system (Beijing, China), was used as the liquid, and the smooth, flat surface of each membrane block was prepared by placing a small rectangle of the membrane on a cleaned glass slide. A needle attached to a micro-syringe was used to dispense droplets of ultrapure water at a rate of one microliter per second onto the surfaces of the membranes. The static contact-angle measurement was measured (Kino Co., Ltd., London, United Kingdom) and obtained from a picture using video-enhanced image processing. To reduce experimental error, the contact angle was measured at least twice randomly for each sample, and the average was reported (Table 3).

### 3.6. Fouling Studies

The investigation into fouling was evaluated by testing the membranes’ permeability and rejection properties using dead-end technology connected to a nitrogen gas cylinder with an active surface area of 7.065 × 10^−4^ m^2^ at a pressure of 0.5 MPa. Each membrane was compacted under a pressure of ultrapure water for at least 30 min, and the mass of pure water rejection was calculated using an analytical balance to weigh the grams of each water rejection every 5 min. The total for 30 min was taken as the measurement, and each sample measurement was repeated once. The permeation flux was calculated using the following equation:(1)$$J=\frac{V}{A\times t}$$
where *V* is the volume of permeate clean water in liters (*L*), *A* is the effective area of the membrane in square meters (m^2^), and *t* is the amount of time it takes for the water to pass through the membrane (hours). A dead-end cell was used for testing the BSA rejection performance of the flat-sheet membranes under the same operating conditions. The BSA solution was prepared by dissolving 0.5 g of BSA in 100 mL of distilled water. The rejection of BSA was calculated using the following equation:(2)$$R=(1-\frac{C_{p}}{C_{f}})\times 1$$

In the above Equation (2), *C_p_* is the concentration of the solute in the permeate, and *C_f_* is the concentration of the solute in the feed stream [41].

## 4. Conclusions

In summary, the phase-inversion technique was used to create PES origin and modified ultra-filtration membranes by incorporating CMAT particles into the PES solutions. The CMAT particles were characterized using various analysis techniques, such as XRD, HRTEM, SEM, and XPS. The XRD analysis showed that the synthesis method successfully produced CMAT, while HRTEM revealed a lattice spacing of 0.25 nm, consistent with the XRD patterns. SEM and XPS characterizations confirmed the presence of all the elements and a uniform distribution of particles with a size less than 5 nm. The contact angle of CMAT was found to be 100% hydrophilic.

For the characterization of the membrane, the SEM results showed that the addition of CMAT did not significantly impact the top layer and cross-section of the PES membranes, and the pores remained normal. The AFM results revealed that the PES membranes with low concentrations of CMAT had a smooth surface, while those with high concentrations of CMAT showed a rough surface. The TGA curves showed little difference between the PES origin membranes and the blended PES membranes regarding temperature losses. The contact-angle results showed that the modified membranes were more hydrophilic than the PES origin membrane due to the addition of CMAT. The PES origin membrane performed better than the membranes mixed with CMAT in terms of water permeability, while the modified membranes with CMAT performed well in the context of fouling resistance.

## Data Availability

Data is available free of charge in MDPI website.

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
