# Peer review of "Preparation and Characterization of Polyethersulfone-Ultrafiltration Membrane Blended with Terbium-Doped Cerium Magnesium Aluminate: Analysis of Fouling Behavior"

_molecules, 2023, doi:10.3390/molecules28062688_

Round 1

Reviewer 1 Report

This manuscript is aimed to prepare and characterized the polyethersulfone (PES) ultrafiltration membrane with terbium-doped cerium magnesium aluminate (CMAT). The materials were characterized with various analytical tools. In my opinion the manuscript in the present form can be accepted after considering the "major revisions" into account. The following points must be considered to accept the revised version:
Comments

1. "Keywords" must be written in good manner.

2. The characterization section should discuss deeply especially the XRD, HRTEM, and SEM analyses.

3. The BET-Surface area should be provided.

4. L-144 to L161 should support by references

5. Fig 6 has poor resolution it should replace and transfer to SI.

6. TGA studies should be discussed with details and support the discussion be references, also the TGA figures is not clear.

7. In section 3.1. Materials; authors should provide the grade of chemicals; i.e., Terbium-doped cerium magnesium aluminate (CMAT), N-Methyl-2-pyrrolidone, Polyvinylpyrrolidone, Polyethersulfone...etc.

8. Chemicals in section 3.2.(L249-252) should be added in materials section.

9. L266 what is the volume used for each component, and also the volume of divided solution

Author Response

Reviewer#1

Comments and Suggestions for Authors

This manuscript is aimed to prepare and characterized the polyethersulfone (PES) ultra-filtration membrane with terbium-doped cerium magnesium aluminate (CMAT). The materials were characterized with various analytical tools. In my opinion the manuscript in the present form can be accepted after considering the "major revisions" into account. The following points must be considered to accept the revised version:

  1. "Keywords" must be written in good manner.

Response1: Thank you for your comments, we have made the possible correction in the keywords and Added in the revised version of manuscrit. Keywords:  Polyethersulfone; ultra-filtration membrane; blending; Ce0.67Tb0.33MgAl11O19; CMAT synthesis.

  1. The characterization section should discuss deeply especially the XRD, HRTEM, and SEM analyses.

Response2:  We made a possible changes in the in the revised version of manuscript interprétation figure 1,2 and 3.

  1. The BET-Surface area should be provided.

Response3: We have added the BET surface area in the revised version of manuscript

  1. L-144 to L161 should support by references.

Response4: The O 1s spectrum envelops was fitted by three peaks. The first peak located at lower binding energy  530.4eV, The second peak, which is almost at the same altitude as 531.8eV and finally, the small peak may be assigned to the oxygen atoms of adsorbed water with characteristic binding energy of approximately 532.5eV attributed to C-O, O-C and O-C correspondingly. Reference is in this article: Enhanced corrosion resistance properties of carbon steel in hydrochloric acid medium by aminotris-(methylenephosphonic): Surface characterizations.

  1. Fig 6 has poor resolution it should replace and transfer to SI.

Response5:  Thank you very much for your suggestion, we have made correction and transfer to SI in the revision version of manuscript.

  1. 6. TGA studies should be discussed with details and support the discussion be references, also the TGA figures is not clear.

Response6 Done: Furthermore in Fig. 8, it shows the typical TGA graphs of the membranes from G0 to G7. we notice a consistency in temperature drop throughout all TGA curves. we observe the membranes G1, G2, G3, G4, G6 and G7 show the same endothermic peak at 80℃ and same exothermic peaks at 40℃ while we see an endothermic peak approximately at 100℃, two particulary exothermic peaks at 40℃ and 100℃ that’s mean for the G5. we can conclude that the TGA membranes reactions starts40℃ except G5 which the reaction starts at 40℃ and finishes at 100℃. The addition of CMAT did not affect a large loss of energy, which means that the slight temperature loss explains C-OH present in the PES membrane surface.

Reference is in this article: Mechanism of CeMgAl11O19: Tb3+ alkaline fusion with sodium hydroxide.

  1. In section 3.1. Materials; authors should provide the grade of chemicals; i.e., Terbium-doped cerium magnesium aluminate (CMAT), N-Methyl-2-pyrrolidone, Polyvinylpyrrolidone, Polyethersulfone...etc.

Response7: we have made the correction and provide the grad of chemicals in in the révise version of manuscript : Terbium-doped cerium magnesium aluminate (CMAT) were purchased from Sigma-Aldrich, CeO2 (Sigma-Aldrich,  211575, powder, <5 μm, 99.9% trace metals basis), Tb4O7 (Sigma-Aldrich, 253952), MgO (Sigma-Aldrich, 342793, ≥99% trace metals basis, -325 mesh) and Al2O3 (Sigma-Aldrich, 229423 powder, 99.99% trace metals basis). N-Methyl-2-pyrrolidone (NMP) reagent used without further purification, Polyethersulfone (PES) ultrason E6020P, Polyvinylpyrrolidone (PVP)  and Bovine serum albumin (BSA, ICN Biomedical) were all bought from Shanghai.        

  1. Chemicals in section 3.2.(L249-252) should be added in materials section.

Response8 : For the synthesis of terbium doped cerium magnesium aluminate (CMAT) with chemical formula of Ce0.67Tb0.33MgAl11O19, the Stoichiometric amounts of  CeO2, Tb4O7, MgO and Al2O3 and have been mixed through a high energy mixer mill 12 hours with zirconia cups and balls (ball to powders ratio was 10:1) in ethanol media at rotation speed of 285 rpm. After mixing process, the obtained mixture has been dried in a drier oven at 80℃. To improve the proper synthesized reaction of CMAT, a pre-shaping process was done on the mixture of powders with uniaxial press at pressure of 120 MPa which led to prepare cylindrical green samples. A sintering process was done on the green bodies with heating rate of 5℃/min at maximum temperature of 1200℃ with 5 hours holding time.

  1. L266 what is the volume used for each component, and also the volume of divided solution.

Response9: Thank you for your suggestion, we have made possible correction in the revised version of manuscript.

Reviewer 2 Report

The manuscript prepared by Aouled et al. demonstrated the performance of PES membrane mixed with different concentrations of CMAT. The authors utilized several material characterization methods to identify experimental samples and finally presented a result of water flux and BSA rejection rate for each sample. This article reads like a report from “Scientific America”, as the authors did not elucidate any scientific issues, but presented data. To address my major concern, please consider following comments.

1.       Does this work focus on materials characterization? If so, you need to consider more thorough discussion about the data. If not, only one figure described the membrane performance is far from enough to tell you story.

2.       You mentioned that biofouling is the major issue, however, BSA is not biofouling, it is organic fouling. So what is your focus?

3.       You title indicates you conducted research to prepare a new membrane by mixing PES with CMAT, then, Introduction indicates your word aimed at solving fouling issue, then Discussion tried to convince the reader you have successfully prepared a new membrane. This is confusing.

4.       All your data lack error bar, which implies your experimental design was flawed.

5.       The only good point of this paper I could think of is to present performances of these membrane samples, so, instead of presenting a Table of Flux and BSA Rejection. Please present dynamic performance results as a function of time (Y axis: Water flux, X axis: Time)

Author Response

Reviewer#2

Comments and Suggestions for Authors

The manuscript prepared by Aouled et al. demonstrated the performance of PES membrane mixed with different concentrations of CMAT. The authors utilized several material characterization methods to identify experimental samples and finally presented a result of water flux and BSA rejection rate for each sample. This article reads like a report from “Scientific America”, as the authors did not elucidate any scientific issues, but presented data. To address my major concern, please consider following comments.

  1. Does this work focus on materials characterization? If so, you need to consider more thorough discussion about the data. If not, only one figure described the membrane performance is far from enough to tell you story.

Response1: This work focus on materials characterization and also the characterization of membranes blended with the terbium doped cerium magnesium aluminate. At the first part we focused on the characterization of the CMAT material that we have done the XRD, XPS, HRTEM, SEM and WCAG to know and i understand that CMAT material contain all the elements are present in the CMAT after characterization. However in the characterization of the membranes, we were focused on the performance of the membranes after mixed with the CMAT.

  1. You mentioned that biofouling is the major issue, however, BSA is not biofouling, it is organic fouling. So what is your focus?

Response2: My focus as i mentioned in the first response was to characterize the CMAT material and membranes mixed with CMAT. As i mentioned the biofouling as a major issue in the introduction it is because i wanted to test the biofouling with some bacteria on the membranes in my research but as i could not prepare it the bacteria in the laboratory because of some issue due to the laboratory so i have finally decided to test the BSA rejection who’s organic fouling instead of biofouling.

  1. You title indicates you conducted research to prepare a new membrane by mixing PES with CMAT, then, Introduction indicates your word aimed at solving fouling issue, then Discussion tried to convince the reader you have successfully prepared a new membrane. This is confusing.

Response3: As i said in the response 2, i wanted to test the biofouling with some bacteria on the membranes but i have decided to test the BSA rejection because of some issues instead of testing the biofouling.

  1. All your data lack error bar, which implies your experimental design was flawed.

Response4: The lack error bar is because it is my first time to write an article and i did not know that without error bar all my experimental design will be flawed.

  1. The only good point of this paper I could think of is to present performances of these membrane samples, so, instead of presenting a Table of Flux and BSA Rejection. Please present dynamic performance results as a function of time (Y axis: Water flux, X axis: Time).

Response5: Thank you for your compliments and suggestion we have made possible correction in the revised version in the manuscript (figure 10a and 10b).

Round 2

Reviewer 1 Report

After check the revised version of manuscript titled “Preparation of Polyethersulfone ultrafiltration membrane blended with terbium-doped cerium magnesium aluminate”. I see the manuscript was improved but still needs more attention especially for the Figure 6 and 8, I suggested in the first round to shift Figure 6 to supplementary materials and improve the resolution of Figure 8. But did not perform. I suggest now to shift both figures to the supplementary materials because not meaningful in the current state with the poor resolution. 

Author Response

Dear Editor,

Thank you very much for your kind consideration of our manuscript. We also express our sincere thanks to the reviewers for the constructive comments. We have revised the manuscript exactly according to the comments. The detailed revisions and answers to the comments are as follows. Highlighted Revised Manuscript (with Yellow Colour) Response to Reviewers and supplementary file are uploaded for review.

Reviewer#1

After check the revised version of manuscript titled “Preparation of Polyethersulfone ultrafiltration membrane blended with terbium-doped cerium magnesium aluminate”. I see the manuscript was improved but still needs more attention especially for the Figure 6 and 8, I suggested in the first round to shift Figure 6 to supplementary materials and improve the resolution of Figure 8. But did not perform. I suggest now to shift both figures to the supplementary materials because not meaningful in the current state with the poor resolution.

Answer: After I reviewed my paper, I shifted the both figures to the supplementary materials and I added SEM cross section image to the figure 6 for more understandable for the readers. Uploaded with revised version of manuscript.

Reviewer 2 Report

Reviewer#2

Comments and Suggestions for Authors

The manuscript prepared by Aouled et al. demonstrated the performance of PES membrane mixed with different concentrations of CMAT. The authors utilized several material characterization methods to identify experimental samples and finally presented a result of water flux and BSA rejection rate for each sample. This article reads like a report from “Scientific America”, as the authors did not elucidate any scientific issues, but presented data. To address my major concern, please consider following comments.

General comments: First, I strongly suggest the author check grammar before sending our official response. Your response contains grammar mistakes that should not appear. Second, you failed to address comments via improper arguing. For example, you supposed to do biofouling but due to your problems, you chose BSA. Doesn’t this mean you need to revise your manuscript? In your comment, you argued “i wanted to test the biofouling with some bacteria on the membranes but i have decided to test the BSA rejection because of some issues instead of testing the biofouling.” Therefore, your response is insufficient to have your manuscript published.

  1. Does this work focus on materials characterization? If so, you need to consider more thorough discussion about the data. If not, only one figure described the membrane performance is far from enough to tell you story.

Response1: This work focus on materials characterization and also the characterization of membranes blended with the terbium doped cerium magnesium aluminate. At the first part we focused on the characterization of the CMAT material that we have done the XRD, XPS, HRTEM, SEM and WCAG to know and i understand that CMAT material contain all the elements are present in the CMAT after characterization. However in the characterization of the membranes, we were focused on the performance of the membranes after mixed with the CMAT.

  1. You mentioned that biofouling is the major issue, however, BSA is not biofouling, it is organic fouling. So what is your focus?

Response2: My focus as i mentioned in the first response was to characterize the CMAT material and membranes mixed with CMAT. As i mentioned the biofouling as a major issue in the introduction it is because i wanted to test the biofouling with some bacteria on the membranes in my research but as i could not prepare it the bacteria in the laboratory because of some issue due to the laboratory so i have finally decided to test the BSA rejection who’s organic fouling instead of biofouling.

Comment: Please revise corresponding part.

  1. You title indicates you conducted research to prepare a new membrane by mixing PES with CMAT, then, Introduction indicates your word aimed at solving fouling issue, then Discussion tried to convince the reader you have successfully prepared a new membrane. This is confusing.

Response3: As i said in the response 2, i wanted to test the biofouling with some bacteria on the membranes but i have decided to test the BSA rejection because of some issues instead of testing the biofouling.

Comment: See previous comment.

  1. All your data lack error bar, which implies your experimental design was flawed.

Response4: The lack error bar is because it is my first time to write an article and i did not know that without error bar all my experimental design will be flawed.

Comment: Error bar is a standard format to present data. You should discuss this with your advisor. In your revised manuscript, you still did not present error bar. I will reject your manuscript if you fail to do so.

  1. The only good point of this paper I could think of is to present performances of these membrane samples, so, instead of presenting a Table of Flux and BSA Rejection. Please present dynamic performance results as a function of time (Y axis: Water flux, X axis: Time).

Comment: I don’t understand your figure, please refer to good articles to see how others present dynamic performance of membranes.

Response5: Thank you for your compliments and suggestion we have made possible correction in the revised version in the manuscript (figure 10a and 10b).

Author Response

Reviewer#2

Comments and Suggestions for Authors

The manuscript prepared by Aouled et al. demonstrated the performance of PES membrane mixed with different concentrations of CMAT. The authors utilized several material characterization methods to identify experimental samples and finally presented a result of water flux and BSA rejection rate for each sample. This article reads like a report from “Scientific America”, as the authors did not elucidate any scientific issues, but presented data. To address my major concern, please consider following comments.

General comments: The grammar have been checked and well-made correction of grammar.

  1. Does this work focus on materials characterization? If so, you need to consider more thorough discussion about the data. If not, only one figure described the membrane performance is far from enough to tell you story.

Response 1: This work focus on the membrane’s characterization and the analysis of fouling behavior of the membranes. I was revised my paper and gave a good title concerning about my work.

  1. You mentioned that biofouling is the major issue, however, BSA is not biofouling, it is organic fouling. So, what is your focus?

Response 2: Now after reviewed my work I focus the study fouling of the membranes for using BSA which is an organic fouling.

  1. You title indicates you conducted research to prepare a new membrane by mixing PES with CMAT, then, Introduction indicates your word aimed at solving fouling issue, then Discussion tried to convince the reader you have successfully prepared a new membrane. This is confusing.

Response 3: The title has been changed after a long review of my article, even I have made some review in the introduction and wrote properly the introduction. Now hope the title and the work I have done are very convenient and understandable for readers.

  1. All your data lack error bar, which implies your experimental design was flawed.

Response 4: Dear reviewer thank you for your kind consideration, the experiments have been repeated and reviewed and error bars are added in all experimental data.

  1. The only good point of this paper I could think of is to present performances of these membrane samples, so, instead of presenting a Table of Flux and BSA Rejection. Please present dynamic performance results as a function of time (Y axis: Water flux, X axis: Time).

Response 5: Thank you for attention, we have revised the paper and added the (Y axis: Water flux, X axis: Time) in the revised version of manuscript.